# Serum Chloride Levels at Hospital Discharge and One-Year Mortality among Hospitalized Patients

**DOI:** 10.3390/medsci8020022

**Published:** 2020-05-19

**Authors:** Tananchai Petnak, Charat Thongprayoon, Wisit Cheungpasitporn, Tarun Bathini, Saraschandra Vallabhajosyula, Api Chewcharat, Kianoush Kashani

**Affiliations:** 1Division of Pulmonary and Critical Care Medicine, Department of Medicine, Mayo Clinic, Rochester, MN 55902, USA; petnak@yahoo.com; 2Division of Pulmonary and Critical Care Medicine, Faculty of Medicine Ramathibodi Hospital, Mahidol University, Bangkok 10100, Thailand; 3Division of Nephrology and Hypertension, Department of Medicine, Mayo Clinic, Rochester, MN 55905, USA; api.che@hotmail.com; 4Division of Nephrology, Department of Internal Medicine, University of Mississippi Medical Center, Jackson, MS 39216, USA; 5Department of Internal Medicine, University of Arizona, Tucson, AZ 85721, USA; tarunjacobb@gmail.com; 6Department of Cardiovascular Medicine, Mayo Clinic, Rochester, MN 55905, USA; Vallabhajosyula.Saraschandra@mayo.edu

**Keywords:** chloride, hypochloremia, hyperchloremia, electrolytes, outcomes, hospitalization, hospital discharge, nephrology

## Abstract

This study aimed to assess the one-year mortality risk based on discharge serum chloride among the hospital survivors. We analyzed a cohort of adult hospital survivors at a tertiary referral hospital from 2011 through 2013. We categorized discharge serum chloride; ≤96, 97–99, 100–102, 103–105, 106–108, and ≥109 mmoL/L. We performed Cox proportional hazard analysis to assess the association of discharge serum chloride with one-year mortality after hospital discharge, using discharge serum chloride of 103–105 mmoL/L as the reference group. Of 56,907 eligible patients, 9%, 14%, 26%, 28%, 16%, and 7% of patients had discharge serum chloride of ≤96, 97–99, 100–102, 103–105, 106–108, and ≥109 mmoL/L, respectively. We observed a U-shaped association of discharge serum chloride with one-year mortality, with nadir mortality associated with discharge serum chloride of 103–105 mmoL/L. When adjusting for potential confounders, including discharge serum sodium, discharge serum bicarbonate, and admission serum chloride, one-year mortality was significantly higher in both discharge serum chloride ≤99 hazard ratio (HR): 1.45 and 1.94 for discharge serum chloride of 97–99 and ≤96 mmoL/L, respectively; *p* < 0.001) and ≥109 mmoL/L (HR: 1.41; *p* < 0.001), compared with discharge serum chloride of 103–105 mmoL/L. The mortality risk did not differ when discharge serum chloride ranged from 100 to 108 mmoL/L. Of note, there was a significant interaction between admission and discharge serum chloride on one-year mortality. Serum chloride at hospital discharge in the optimal range of 100–108 mmoL/L predicted the favorable survival outcome. Both hypochloremia and hyperchloremia at discharge were associated with increased risk of one-year mortality, independent of admission serum chloride, discharge serum sodium, and serum bicarbonate.

## 1. Introduction

Chloride ions are the principal extracellular strong anions in humans [1,2]. Chloride is substantially involved in the maintenance of acid-base balance, plasma osmotic pressure, hydrochloric acid production, osmotic gradient in the gastrointestinal tract, kidney function, and electrolyte activity in the cellular function [2]. The abnormality of serum chloride levels, or dyschloremia, is associated with several adverse consequences. Previous studies demonstrated that hyperchloremia could result in kidney vasoconstriction, leading to decreased renal blood flow and glomerular filtration rate (GFR). In contrast, hypochloremia might enhance inflammatory response through increased cytokine release [3,4,5,6,7].

Although serum chloride is the major anion commonly reported in the basic metabolic panel, its importance is often overlooked by physicians. The incidence of dyschloremia ranges from 25% to 43% [8,9,10,11]. The causes of hyperchloremia include the loss of bicarbonate ions (either by kidney or gastrointestinal loss), hemodilution, and the infusion of chloride-rich fluids [2]. The causes of hypochloremia include chloride loss by either kidney (e.g., diuretic use), or gastrointestinal loss (e.g., vomiting) [2].

The reports related to the impact of serum chloride levels on patient outcomes have been mostly limited to the field of critical care, and the majority of the previous studies focused on the impact of admission serum chloride on the hospital outcomes [8,9,10,11,12,13,14,15,16]. The knowledge of the impact of serum chloride levels at hospital discharge on long-term mortality after hospital discharge remains limited. Therefore, we conducted this study to assess the risk of one-year mortality based on discharge serum chloride levels in adult hospitalized patients.

## 2. Materials and Methods

### 2.1. Study Population

This is a single-center cohort study conducted at Mayo Clinic Hospital, Minnesota, USA. We screened all hospitalized adult patients who survived until hospital discharge from 2011 to 2013. We included patients who had at least two serum chloride measurements during hospitalization, of which, one of serum chloride was measured within 24 h of hospital admission to represent admission serum chloride level. We included only the first hospital admission for patients with multiple hospital admissions during the study period. Mayo Clinic institutional review board approved this study (IRB number 15-000024) and waived the need for informed consent due to the minimal risk nature of the study.

### 2.2. Data Collection

We assessed discharge serum chloride as the primary predictor of one-year mortality after hospital discharge. Discharge serum chloride was the last serum chloride measured during hospitalization, while admission serum chloride was the first serum chloride measured during the first 24 h of hospitalization. We calculated estimated glomerular filtration (eGFR) using the Chronic Kidney Disease Epidemiology Collaboration equation [17]. We classified principal diagnoses based on International Classification of Disease, Ninth Revision (ICD-9) codes. We computed the Charlson Comorbidity Index to assess the comorbidity burden of individual patients. We determined patients’ vital status from our institutional registry and the Social Security Death Index database.

### 2.3. Statistical Analysis

We described continuous variables using mean ± standard deviation (SD), and categorical variables using frequency with percentages. We categorized discharge serum chloride into six groups; ≤96, 97–99, 100–102, 103–105, 106–108, and ≥109 mmoL/L based on its distribution at 10th, 25th, 50th, 75th, 90th percentile to evaluate the non-linear association with mortality. We constructed the restricted cubic spline with 4 knots at 96, 100, 104, and 108 mmoL/L of serum chloride to depict the non-linear association between serum chloride and one-year mortality. We compared continuous variables using ANOVA and categorical variables using the Chi-square test. We measured patient survival from hospital discharge and followed until death or one year after hospital discharge. For patients who were lost to follow-up or whose vital status was not known, they were censored at the date of their last inpatient/outpatient follow-up visit. We estimated one-year mortality risk using the Kaplan–Meier method and compared one-year mortality risk among discharge serum chloride groups using the log-rank test. We performed Cox proportional hazard analysis to calculate the hazard ratio (HR) of one-year mortality after hospital discharge for various discharge serum chloride groups using discharge serum chloride of 103–105 mmoL/L as the reference group as it was associated with the nadir mortality at one year. We adjusted multivariable analysis for baseline characteristics in Table 1 that significantly differed among discharge serum chloride groups (*p* < 0.05). We performed a stratified analysis based on admission serum chloride (≤99, 100–108, and ≥109 mmoL/L). We performed an interaction test to assess if the association of discharge serum chloride with mortality differed by admission serum chloride by adding the interaction term in the multivariable model. A two-tailed *p*-value of less than 0.05 was considered statistically significant. All analyses were performed using JMP statistical software (version 14.0, SAS Institute, Cary, NC, USA, 2012).

## 3. Results

### 3.1. Clinical Characteristics

We studied a total of 56,907 hospitalized patients who were discharged from our hospital. The mean admission serum chloride was 103 ± 5 mmoL/L. The mean discharge serum chloride was 102 ± 4 mmoL/L. Discharge serum chloride of ≤96, 97–99, 100–102, 103–105, 106–108, and ≥109 mmoL/L was found in 9%, 14%, 26%, 28%, 16%, and 7% of patients, respectively. Table 1 shows the clinical characteristics based on the discharge serum chloride levels.

### 3.2. Discharge Serum Chloride and One-Year Mortality

The median follow-up time was 464 (interquartile range 112–914) days. Of 56,907 patients, 6447 (11.3%) died within one year after hospital discharge. The restricted cubic spline (Figure 1) demonstrates a U-shaped association of discharge serum chloride with one-year mortality, with the nadir mortality associate with discharge serum chloride of 103–105 mmoL/L.

Kaplan–Meier plot (Figure 2) indicates an estimated one-year mortality of 24.2% in discharge serum chloride of ≤96 mmoL/L, 17.0% in 97–99 mmoL/L, 12.7% in 100–102 mmoL/L, 11.0% in 103–105 mmoL/L, 11.8% in 106–108 mmoL/L, and 17.9% in ≥109 mmoL/L (*p* < 0.001; Table 2).

When adjusting for age, sex, race, eGFR, principal diagnosis, comorbidities, in-hospital clinical events and treatments, discharge serum sodium and serum bicarbonate, and admission serum chloride, the risk of one-year mortality in hypochloremic patients was higher when compared with discharge serum chloride of 103–105 mmoL/L for discharge serum chloride levels of 97–99 mmoL/L and ≤96 mmoL/L (hazard ratio (HR): 1.45, 95% confidence interval (CI): 1.32–1.59; and HR: 1.94, 95% CI: 1.72–2.19, respectively). Discharge hyperchloremia of ≥109 mmoL/L was also significantly associated with increased risk of one-year mortality (HR: 1.41 (95% CI: 1.26–1.58), compared with discharge serum chloride of 103–105 mmoL/L; Table 2).

### 3.3. Stratified Analysis Based on Admission Serum Chloride

In the subgroup analysis of patients with admission serum chloride of ≤99, and 100–108 mmoL/L, both discharge hypochloremia of ≤99 mmoL/L and discharge hyperchloremia of ≥109 mmoL/L were significantly associated with increased risk of one-year mortality when compared with discharge euchloremia of 100–108 mmoL/L (Table 3). In contrast, in the subgroup analysis of patients with admission hyperchloremia of ≥109 mmoL/L, discharge hypochloremia of ≤99 mmoL/L and discharge hyperchloremia of ≥109 mmoL/L were not significantly associated with increased risk of one-year mortality, compared with discharge euchloremia (i.e., 100–108 mmoL/L) (Table 3). We found an interaction between admission and discharge serum chloride on one-year mortality (interaction *p* = 0.002).

## 4. Discussion

The serum chloride level has often been overlooked in practice. Most of the studies generally highlighted the role of serum chloride levels either at the admission or the alteration of serum chloride levels during hospitalization. To our best knowledge, this study is the first study focusing on the impact of the discharge serum chloride levels on the one-year mortality. This large cohort demonstrated a U-shaped association between discharge serum chloride levels and one-year mortality, with the optimal discharge serum chloride levels ranging from 100–108 mmoL/L. The deviation of discharge serum chloride levels from this range resulted in a progressive increase in one-year mortality. In patients with admission serum chloride of ≤108 mmoL/L, both hypochloremia and hyperchloremia at discharge were associated with higher one-year mortality, whereas, in patients with admission serum chloride of ≥109 mmoL/L, none of dyschloremia at discharge was significantly associated with higher one-year mortality.

Several studies previously reported the impact of serum chloride levels on mortality. These investigations demonstrated the effect of hyperchloremia on outcomes, mostly in critically ill patients. Both admission and hospital-acquired hyperchloremia were associated with worse patient outcomes, including a higher risk of acute kidney injury, mortality, and longer length of hospital stay [9,12,13,14,16,18,19,20]. The increase in serum chloride levels out of the proportion to serum sodium levels results in the reduction of strong ion difference, leading to hyperchloremic metabolic acidosis [2]. The severity of acidosis might be enhanced when the hyperchloremic metabolic acidosis is accompanied by anion gap metabolic acidosis.

Our study also demonstrated the association of hypochloremia at discharge with one-year mortality. In contrast to hyperchloremia, the impact of hypochloremia on patient outcomes has not been extensively studied. Only a few investigations demonstrated that hypochloremia was associated with increased adverse outcomes in critically ill patients [8,10]. Besides, hypochloremia was associated with higher mortality in patients with specific conditions, including chronic kidney disease, cardiovascular disease, and acute ischemic stroke [21,22,23,24,25]. Since serum chloride levels, particularly in cardiovascular disease, may indicate the severity of these diseases, hypochloremia might represent a higher severity of illness, resulting in a greater risk of mortality [2]. Also, metabolic alkalosis induced by hypochloremia can cause hypercapnic respiratory acidosis in chronic obstructive pulmonary disease patients, resulting in a higher need for ventilatory support [15]. In the present study, approximately 40% of patients with discharge serum chloride levels <100 mmoL/L consisted of patients with cardiovascular disease, stroke, and chronic obstructive pulmonary disease. The high proportion of these patients might have magnified the impact of hypochloremia in our study.

In the subgroup analysis, among patients with admission serum chloride levels ≤108 mmoL/L, both hypochloremia and hyperchloremia at discharge were associated with the increased risk of one-year mortality. In contrast, among patients with admission serum chloride ≥109 mmoL/L, hypochloremia and hyperchloremia at discharge were not significantly associated with increased risk of one-year mortality. A previous study demonstrated that the alteration of serum chloride during hospitalization was associated with increased mortality [26]. The upward trend of serum chloride change was significantly associated with increased mortality, whereas the downward trend of serum chloride change was not consistently associated with increased mortality.

This study has some limitations. This was a single-center cohort study. Thus, we cannot establish a causal relationship between discharge serum chloride and mortality. Although we extensively adjusted for known potential confounders, the association might remain unadjusted for unmeasured or unknown factors, such as the causes of dyschloremia, the severity of the underlying disease, the causes of death, the use of intravenous fluid, medications, acid-base status, and the change in serum chloride during hospitalization. Besides, our study population included patients in the white race predominantly and this might limit the generalizability of the result.

## 5. Conclusions

In conclusion, we demonstrated the U-shape association of discharge serum chloride levels with one-year mortality. The optimal range of discharge serum chloride was from 100–108 mmoL/L, as this range was associated with the nadir of one-year mortality. Both hypochloremia and hyperchloremia at discharge were associated with increased one-year mortality. Future trials are needed to assess the impact of protocols to correct dyschloremia before discharge on patients’ outcomes.

## Figures and Tables

**Figure 1 medsci-08-00022-f001:**
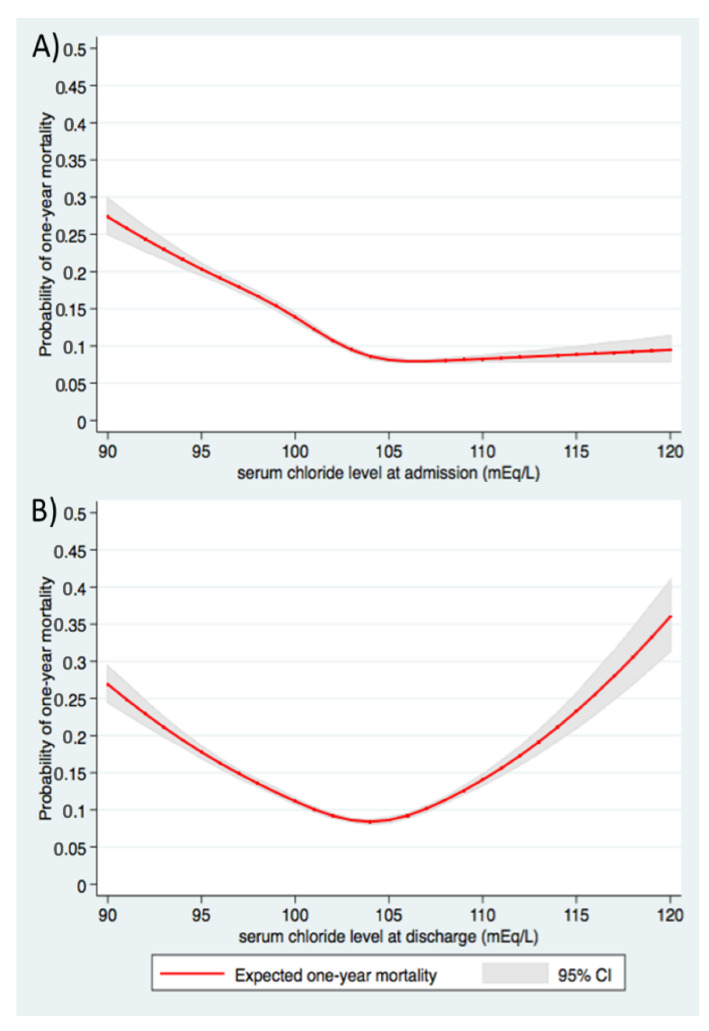
U-shaped association between one-year mortality and (**A**) admission serum chloride and (**B**) discharge serum chloride. Abbreviation: CI, confidence interval.

**Figure 2 medsci-08-00022-f002:**
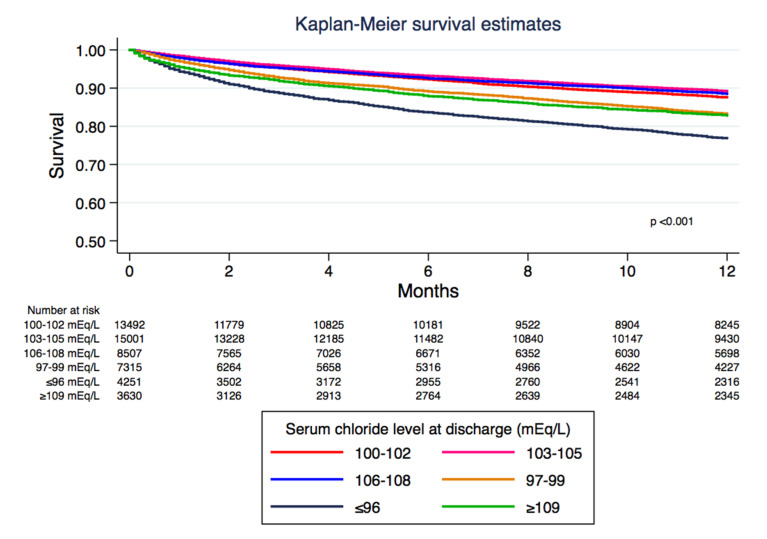
The patient survival based on discharge serum chloride levels.

**Table 1 medsci-08-00022-t001:** Baseline clinical characteristics.

Variables	All	Discharge Serum Chloride Level (mmoL/L)						*p*-Value
		≤96	97–99	100–102	103–105	106–108	≥109	
*N*	56,907	5065	8017	14,576	16,204	9129	3916	
Age (year)	63 ± 17	67 ± 16	64 ± 17	62 ± 17	62 ± 17	63 ± 18	63 ± 18	<0.001
Male	30,447 (54)	2675 (53)	4646 (58)	8319 (57)	8689 (54)	4398 (48)	1720 (44)	<0.001
Caucasian	52,947 (93)	4736 (94)	7493 (93)	13,495 (93)	15,096 (93)	8509 (93)	3618 (92)	0.04
eGFR (mL/min/1.73 m^2^)	79 ± 28	72 ± 31	77 ± 30	81 ± 27	81 ± 27	79 ± 28	72 ± 31	<0.001
Principal Diagnosis								<0.001
- Cardiovascular	14,537 (26)	1713 (34)	2417 (30)	4051 (28)	4056 (25)	1803 (20)	497 (13)	
- Endocrine/Metabolic	1677 (3)	224 (4)	212 (3)	354 (2)	411 (3)	294 (3)	182 (5)	
- Gastrointestinal	5886 (10)	384 (8)	667 (8)	1276 (9)	1638 (10)	1191 (13)	730 (19)	
- Hematology/Oncology	7686 (14)	652 (13)	1218 (15)	2062 (14)	2158 (13)	1147 (13)	449 (11)	
- Infectious Disease	2097 (4)	155 (3)	230 (3)	471 (3)	532 (3)	439 (5)	270 (7)	
- Respiratory	2595 (5)	369 (7)	469 (6)	632 (4)	662 (4)	307 (3)	156 (4)	
- Genitourinary	1854 (3)	119 (2)	191 (2)	334 (2)	492 (3)	393 (4)	325 (8)	
- Injury/poisoning	8737 (15)	644 (13)	1163 (15)	2308 (16)	2517 (16)	1472 (16)	633 (16)	
- Other	11,838 (21)	805 (16)	1450 (18)	3088 (21)	3738 (23)	2083 (23)	674 (17)	
Charlson Score	1.9 ± 2.4	2.4 ± 2.6	2.1 ± 2.5	1.9 ± 2.4	1.8 ± 2.3	1.9 ± 2.4	2.2 ± 2.5	<0.001
Comorbidities								
- Coronary artery disease	4788 (8)	566 (11)	715 (9)	1200 (8)	1274 (8)	724 (8)	309 (8)	<0.001
- Congestive heart failure	4648 (8)	792 (16)	874 (11)	1158 (8)	1024 (6)	528 (6)	272 (7)	<0.001
- Peripheral vascular disease	2042 (4)	268 (5)	352 (4)	513 (4)	460 (3)	302 (3)	147 (4)	<0.001
- Stroke	4649 (8)	516 (10)	630 (8)	1121 (8)	1271 (8)	764 (8)	347 (9)	<0.001
- Diabetes mellitus	12,438 (22)	1469 (29)	2018 (25)	3319 (23)	3088 (19)	1695 (19)	849 (22)	<0.001
- Chronic obstructive pulmonary disease	5453 (10)	832 (16)	1016 (13)	1327 (9)	1300 (8)	672 (7)	306 (8)	<0.001
- Cirrhosis	1632 (3)	171 (3)	219 (2)	373 (3)	371 (2)	303 (3)	195 (5)	<0.001
Acute myocardial infarction	2762 (5)	237 (5)	342 (4)	729 (5)	887 (5)	447 (5)	120 (3)	<0.001
Acute kidney injury	8420 (15)	1316 (26)	1552 (19)	2106 (14)	1907 (12)	998 (11)	541 (14)	<0.001
Renal replacement therapy	1180 (2)	379 (7)	325 (4)	248 (2)	133 (1)	51 (1)	44 (1)	<0.001
Mechanical ventilation	9418 (17)	1250 (25)	1933 (24)	2764 (19)	2209 (14)	921 (10)	341 (9)	<0.001
Vasopressor use	5282 (9)	832 (16)	1113 (14)	1496 (10)	1147 (7)	485 (5)	209 (5)	<0.001
Admission serum chloride (mmoL/L)	103 ± 5	98 ± 7	101 ± 5	103 ± 5	104 ± 4	105 ± 4	107 ± 5	<0.001
Discharge serum sodium (mmoL/L)	138 ± 3	134 ± 4	136 ± 3	138 ± 2	139 ± 2	141 ± 2	142 ± 3	<0.001
Discharge serum bicarbonate (mmoL/L)	26 ± 3	29 ± 4	28 ± 3	27 ± 3	26 ± 3	25 ± 3	22 ± 3	<0.001

Continuous data are presented as mean ± standard deviation (SD); categorical data are presented as count (%); Abbreviation: eGFR, estimated glomerular filtration rate.

**Table 2 medsci-08-00022-t002:** The association between discharge serum chloride and one-year mortality.

Discharge Serum Chloride (mmoL/L)	One-Year Mortality	Univariate Analysis		Multivariate Analysis	
		HR (95% CI)	*p*	Adjusted HR * (95% CI)	*p*
≤96	24.2%	2.46 (2.26–2.66)	<0.001	1.94 (1.72–2.19)	<0.001
97–99	17.0%	1.61 (1.49–1.75)	<0.001	1.45 (1.32–1.59)	<0.001
100–102	12.7%	1.17 (1.08–1.25)	<0.001	1.07 (0.99–1.15)	0.09
103–105	11.0%	1 (ref)	-	1 (ref)	-
106–108	11.8%	1.09 (0.99–1.18)	0.06	1.01 (0.93–1.10)	0.81
≥109	17.9%	1.75 (1.59–1.92)	<0.001	1.41 (1.26–1.58)	<0.001

* Adjusted for age, sex, race, eGFR, principal diagnosis, Charlson comorbidity score, coronary artery disease, congestive heart failure, peripheral vascular disease, stroke, diabetes mellitus, chronic obstructive pulmonary disease, cirrhosis, acute myocardial infarction, acute kidney injury, renal replacement therapy, mechanical ventilation, vasopressor use, discharge serum sodium, discharge serum bicarbonate, and admission serum chloride. Abbreviation: CI, confidence interval; HR, hazard ratio

**Table 3 medsci-08-00022-t003:** Subgroup analysis based on admission serum chloride.

Discharge Serum Chloride (mEq/L)	One-Year Mortality (%)	Univariate Analysis		Multivariate Analysis	
		HR (95% CI)	*p*	Adjusted HR * (95% CI)	*p*
Admission serum chloride ≤99 mmoL/L					
≤99	26.4%	1.50 (1.38–1.64)	<0.001	1.25 (1.11–1.41)	<0.001
100–108	18.6%	1 (ref)	-	1 (ref)	-
≥109	37.5%	2.31 (1.82–2.92)	<0.001	2.06 (1.60–2.66)	<0.001
Admission serum chloride 100–108 mmoL/L					
≤99	15.1%	1.43 (1.32–1.55)	<0.001	1.37 (1.23–1.52)	<0.001
100–108	10.9%	1 (ref)	-	1 (ref)	-
≥109	16.1%	1.58 (1.41–1.77)	<0.001	1.24 (1.09–1.42)	0.002
Admission serum chloride ≥109 mmoL/L					
≤99	8.9%	0.90 (0.67–1.17)	0.45	1.32 (0.94–1.84)	0.10
100–108	9.8%	1 (ref)	-	1 (ref)	-
≥109	17.6%	1.94 (1.62–2.32)	<0.001	1.10 (0.86–1.40)	0.46

* Adjusted for age, sex, race, eGFR, principal diagnosis, Charlson comorbidity score, coronary artery disease, congestive heart failure, peripheral vascular disease, stroke, diabetes mellitus, chronic obstructive pulmonary disease, cirrhosis, acute myocardial infarction, acute kidney injury, renal replacement therapy, mechanical ventilation, vasopressor use, discharge serum sodium, discharge serum bicarbonate, and admission serum chloride. Abbreviation: CI, confidence interval; HR, hazard ratio

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
