# Peer review of "Serum Chloride Levels at Hospital Discharge and One-Year Mortality among Hospitalized Patients"

_medsci, 2020, doi:10.3390/medsci8020022_

Round 1

Reviewer 1 Report

In this manuscript, Petnak and colleagues investigated the association between serum chloride level at discharge and one-year mortality, in a single center basis- retrospective observational study. They reported that the optimal range of serum chloride levels at discharge as 100-108 mmol/L, with the lowest observed mortality. Although this study interestingly approached the frequently overlooked anion, chloride, I would raise several concerns as follows: 

Major: 

The present study defined serum chloride levels at hospital discharge as a blood test result within 48 hours before discharge. There was no description of how many patients were excluded because of a lack of the chloride data. The fact that this study included only those with chloride test within 48 hours before discharge limits the generalizability of the results. 

The authors applied restricted cubic spline model for generating Figure 1, which is not explained in Methods section. Cubic spline model usually extrapolates the result between the actual mortality values of categorized groups. Any assumption (number of knots, degree of the used function, etc.) for the analysis should be clarified. It is also questionable whether there was enough sample size for the marginal area of the Figure, i.e. chloride level of around 90 or 120 mmol/L. The result of the association between admission chloride level and one-year mortality in this cohort would also highlight the result of Figure 1, in the context of previous studies.

The criteria for predictors selection in multivariate Cox hazard analysis is unclear. They did not include net difference of serum chloride level between admission and discharge. Serum bicarbonate level, which may be associated with the difference between sodium and chloride, would also affect the outcome. 

There is no description of the follow-up rate in each patient group. Number-at-risk can be presented in the Kaplan-Meier method result in Figure 2. 

Minors: 

The result of Log-rank test is not presented in the main text and Figure 2. 
Table 2 does not include “admission” chloride data. Its title should be corrected. 
A more detailed description on the method for interaction analysis would be helpful. It was unclear whether the authors added interaction terms in the model and validate their effects.

Author Response

Response to Reviewer#1

In this manuscript, Petnak and colleagues investigated the association between serum chloride level at discharge and one-year mortality, in a single center basis- retrospective observational study. They reported that the optimal range of serum chloride levels at discharge as 100-108 mmol/L, with the lowest observed mortality. Although this study interestingly approached the frequently overlooked anion, chloride, I would raise several concerns as follows:

Response: We thank you for reviewing our manuscript and for your critical evaluation.

Comment #1

The present study defined serum chloride levels at hospital discharge as a blood test result within 48 hours before discharge. There was no description of how many patients were excluded because of a lack of the chloride data. The fact that this study included only those with chloride test within 48 hours before discharge limits the generalizability of the results.

Response: Thank you. We excluded 1,957 patients because they had no measured serum chloride measurement within 48 hours before discharge. Due to limited generalizability of the result, as pointed out by the reviewer, we revised our inclusion criteria and included those 1957 patients, who were previously excluded because of no serum chloride measurement within 48 hours, in the analysis. Accordingly, the new total number of patients included in the analysis was 56,907.

Comment #2

The authors applied restricted cubic spline model for generating Figure 1, which is not explained in Methods section. Cubic spline model usually extrapolates the result between the actual mortality values of categorized groups. Any assumption (number of knots, degree of the used function, etc.) for the analysis should be clarified. It is also questionable whether there was enough sample size for the marginal area of the Figure, i.e. chloride level of around 90 or 120 mmol/L. The result of the association between admission chloride level and one-year mortality in this cohort would also highlight the result of Figure 1, in the context of previous studies.

Response: This is a very important point. Thank you. The following statements have been added to the statistical analysis section to describe restricted cubic spline construction.

"We constructed the restricted cubic spline with 4 knots at 96, 100, 104, and 108 mmol/L of serum chloride to depict the non-linear association between serum chloride and one-year mortality."

We limited the range of serum chloride in restricted cubic spline from 90 to 120 mmol/L as they represent 1st and 99th percentile. As shown in Figure 1, the 95% confidence interval in the marginal range are acceptable.

We also constructed the restricted cubic spline for the association between admission serum chloride and one-year mortality, as suggested, and included it in Figure 1.

Comment #3

The criteria for predictors selection in multivariate Cox hazard analysis is unclear. They did not include net difference of serum chloride level between admission and discharge. Serum bicarbonate level, which may be associated with the difference between sodium and chloride, would also affect the outcome.

Response: Thank you. The following statements have been added to the statistical analysis section to describe covariate selection in multivariable Cox hazard analysis.

"We adjusted multivariable analysis for baseline characteristics in Table 1 that significantly differed among discharge serum chloride groups (p<0.05)."

We additionally adjusted the multivariable analysis for discharge serum bicarbonate, as suggested. However, as we included both admission and discharge serum chloride in the multivariable model, we could not further adjust for the net difference between admission and discharge serum chloride because of their collinearity.

Comment #4

There is no description of the follow-up rate in each patient group. Number-at-risk can be presented in the Kaplan-Meier method result in Figure 2.

Response: Thank you. The following statements have been added to the result section to describe the follow-up time of this cohort.

"The median follow-up time was 464 (IQR 112-914) days."

Besides, the number-at-risk has been presented in the Kaplan-Meier plot in Figure 2, as suggested.

Comment #5

The result of Log-rank test is not presented in the main text and Figure 2.

Response: The result of the Log-rank test has been added to the result section of the main text, and Figure 2, as suggested.

Comment #6

Table 2 does not include "admission" chloride data. Its title should be corrected

Response: We apologize for this error. The title of table 2 has been corrected, as suggested.

Comment #7

A more detailed description on the method for interaction analysis would be helpful. It was unclear whether the authors added interaction terms in the model and validate their effects.

Response: The following statements have been added to the statistical analysis to describe an interaction test.

"We performed an interaction test to assess if the association of discharge serum chloride with mortality differed by admission serum chloride by adding the interaction term in the multivariable model."

Besides, we performed a stratified analysis of discharge serum chloride and one-year morality based on admission serum chloride (≤99, 100-108, and ≥109 mmol/L) to demonstrate this interaction.

We greatly appreciated the editor and reviewer’s time and comments to improve our manuscript.

Reviewer 2 Report

The findings from this paper are excellent and worthy to review.
This manuscript contained some questions described below.
I think this paper has very interesting, this study contributes to future's clinical medicine largely. I have some questions from a point of view of clinical medicine.

This paper is reviewed from a very interesting point of view. I read it interestingly. However, the lack of specificity of data makes it less convincing.

The data are obtained from a large number of patients and the data are considered reliable. However, the survival rate is greatly affected by clinical background, disease, etc., so that the content of this paper is not concrete even if the kidney diseases and cardiovascular events mentioned in the discussion section are high. It is necessary to describe in detail the specific causes of death and the treatment diseases during hospitalization in the vast amount of data examined this time.

Please add this data to the background of kidney diseases, acidosis or alkalosis, and cardiovascular events.

Author Response

Response to Reviewer #2

The findings from this paper are excellent and worthy to review.

This manuscript contained some questions described below.

I think this paper has very interesting, this study contributes to future's clinical medicine largely. I have some questions from a point of view of clinical medicine.

This paper is reviewed from a very interesting point of view. I read it interestingly. However, the lack of specificity of data makes it less convincing.

Response: We thank you for reviewing our manuscript and for your critical evaluation.

Comment #1

The data are obtained from a large number of patients and the data are considered reliable. However, the survival rate is greatly affected by clinical background, disease, etc., so that the content of this paper is not concrete even if the kidney diseases and cardiovascular events mentioned in the discussion section are high. It is necessary to describe in detail the specific causes of death and the treatment diseases during hospitalization in the vast amount of data examined this time.

Please add this data to the background of kidney diseases, acidosis or alkalosis, and cardiovascular events.

Response: We added the data on discharge serum bicarbonate, and in-hospital clinical events and treatments, such as acute myocardial infarction, acute kidney injury, renal replacement therapy, mechanical ventilation, and vasopressor use, in Table 1, and additionally adjusted the multivariable analysis for these variables. However, our database lacks the detail of the causes of death. The following statements have been added to the limitation section.

"Although we extensively adjusted for known potential confounders, the association might remain unadjusted for unmeasured or unknown factors, such as the causes of dyschloremia, the severity of the underlying disease, the causes of death, the use of intravenous fluid, medications, acid-base status, and the change in serum chloride during hospitalization."

We greatly appreciated the editor and reviewer’s time and comments to improve our manuscript.

Round 2

Reviewer 1 Report

The authors sufficiently responded to my comments.

Reviewer 2 Report

This paper is well written and informative.

I have no specific comments.